# The Role of Phylogenetics in Unravelling Patterns of HIV Transmission towards Epidemic Control: The Quebec Experience (2002–2020)

**DOI:** 10.3390/v13081643

**Published:** 2021-08-19

**Authors:** Bluma G. Brenner, Ruxandra-Ilinca Ibanescu, Nathan Osman, Ernesto Cuadra-Foy, Maureen Oliveira, Antoine Chaillon, David Stephens, Isabelle Hardy, Jean-Pierre Routy, Réjean Thomas, Jean-Guy Baril, Roger Leblanc, Cecile Tremblay, Michel Roger

**Affiliations:** 1McGill Centre for Viral Diseases, Lady Davis Institute for Medical Research, Montréal, QC H3T 1E2, Canada; ruxandra-ilinca.ibanescu@mail.mcgill.ca (R.-I.I.); nathan.osman@mail.mcgill.ca (N.O.); ernesto.cuadrafoy@mail.mcgill.ca (E.C.-F.); maureen.oliveira@jgh.mcgill.ca (M.O.); 2Department of Microbiology and Immunology, McGill University, Montréal, QC H4A 3J1, Canada; 3Department of Medicine (Surgery, Infectious Disease), McGill University, Montréal, QC H3A 2M7, Canada; 4Department of Medicine, University of California, San Diego, CA 93903, USA; achaillon@ucsd.edu; 5Department of Mathematics and Statistics, McGill University, Montréal, QC H3A 0B9, Canada; david.stephens@mcgill.ca; 6Département de Microbiologie et d’Immunologie et Centre de Recherche du Centre Hospitalier de l’Université de Montréal (CHUM), Montréal, QC H2X 0C1, Canada; isabelle.hardy.chum@ssss.gouv.qc.ca (I.H.); cecile.tremblay@umontreal.ca (C.T.); michel.roger.chum@ssss.gouv.qc.ca (M.R.); 7Chronic Viral Illness Service, McGill University Health Centre, Montréal, QC H3A 3J1, Canada; jean-pierre.routy@mcgill.ca; 8Clinique Médicale l’Actuel, Montréal, QC H2L 4P9, Canada; rejean.thomas@lactuel.ca; 9Clinique Médicale Urbaine du Quartier Latin, Montréal, QC H2L 4E9, Canada; jgbaril@videtron.ca; 10Clinique Médicale OPUS, Montréal, QC H3A 1T1, Canada; rogerpleblanc@me.com; 11Montreal PHI Cohort of the Réseau Sida et Maladies Infectieuses, Centre de Recherche du CHUM, Montréal, QC H2X 0A9, Canada

**Keywords:** HIV-1 transmission, men having sex with men, non-B subtypes, phylogenetics, HIV-TRACE, treatment-as-prevention, migration

## Abstract

Phylogenetics has been advanced as a structural framework to infer evolving trends in the regional spread of HIV-1 and guide public health interventions. In Quebec, molecular network analyses tracked HIV transmission dynamics from 2002–2020 using MEGA10-Neighbour-joining, HIV-TRACE, and MicrobeTrace methodologies. Phylogenetics revealed three patterns of viral spread among Men having Sex with Men (MSM, *n* = 5024) and heterosexuals (HET, *n* = 1345) harbouring subtype B epidemics as well as B and non-B subtype epidemics (*n* = 1848) introduced through migration. Notably, half of new subtype B infections amongst MSM and HET segregating as solitary transmissions or small cluster networks (2–5 members) declined by 70% from 2006–2020, concomitant to advances in treatment-as-prevention. Nonetheless, subtype B epidemic control amongst MSM was thwarted by the ongoing genesis and expansion of super-spreader large cluster variants leading to micro-epidemics, averaging 49 members/cluster at the end of 2020. The growth of large clusters was related to forward transmission cascades of untreated early-stage infections, younger at-risk populations, more transmissible/replicative-competent strains, and changing demographics. Subtype B and non-B subtype infections introduced through recent migration now surpass the domestic epidemic amongst MSM. Phylodynamics can assist in predicting and responding to active, recurrent, and newly emergent large cluster networks, as well as the cryptic spread of HIV introduced through migration.

## 1. Introduction

The HIV/AIDS pandemic remains a global health challenge with an estimated 38 million persons living with HIV and 1.7–2 million new cases added annually over the last decade [1]. In Western world settings, *concentrated* HIV-1 subtype B epidemics circulate in key vulnerable populations, including Men having Sex with Men (MSM), People Who Inject Drugs (PWID), and marginalized Heterosexual groups [2,3,4]. In Africa and Asia, generalized Heterosexual (HET) epidemics have diversified to include 10 HIV-1 subtypes and over 40 circulating recombinant forms [5]. Human migration has led to changing demographics, wherein non-B subtypes now account for 20–60% of new infections in the Americas and Europe [4,6,7].

Unprecedented advances in antiretroviral therapy has transformed HIV-1 from a deadly disease to a treatable and potentially eradicable pandemic [8]. The goal of treatment has shifted from addressing individual health benefits to population-level control of HIV epidemics. In 2014, the World Health Organization and UNAIDS launched the “90-90-90” and “95-95-95” treatment-as-prevention initiatives to reduce new infections to 500,000 and 200,000 by 2020 and 2030, respectively. By 2020, all nations were called upon to (1) diagnose 90% of HIV-infected persons; (2) treat 90% of those infected; and (3) sustain viral suppression in 90% of those who are treated. Revised guidelines incentivized HIV testing and immediate initiation of antiviral therapy and expanded access to Pre-Exposure Prophylaxis and Post-Exposure Prophylaxis for high-risk populations, e.g., MSM [9].

Despite concerted “90-90-90” responses, efforts to reduce HIV incidence have stalled. While promising 38% declines in new HIV infections have been observed in eastern and southern Africa over the last decade, many countries have not attained measurable progress, with 72% increases in new HIV infections in eastern Europe and central Asia [10]. In Canada, there are an estimated 65,000 living with HIV with a 25% increase in new HIV cases since 2014 [11,12].

Epidemic control is complicated by varying patterns of sexual risk, long durations of infection (decades) and shifting demographics. Molecular network analyses, using novel bioinformatic tools, have been advanced as structural frameworks to infer key determinants implicated in viral spread that cannot be estimated by classical epidemiological approaches [13,14]. Large sequence datasets from drug resistance programs provide in- depth population-level sampling of regional HIV epidemics [4,13,15,16]. The “clustering” of sequences can ascertain population-level dynamics of HIV-1 spread assessed on chronological and stage-of-infection time scales [17,18]. Molecular surveillance can be integrated with epidemiological, demographic, and behavioural data to describe underlying factors contributing to the growth of individual epidemics in time and space.

In February 2019, the Department of Health and Human Services (DHHS) in the United States launched the End the HIV epidemic by 2030 initiative, incorporating phylogenetic network analysis as a fourth pillar alongside test-treat-suppress measures (https://www.hiv.gov/federal-response/ending-the-hiv-epidemic/federal-action/agencies, accessed on 16 August 2021). Molecular surveillance can assist in identifying HIV-1 outbreaks in 46 key rural and urban jurisdictions. The HIV TRAnsmission Cluster Engine (HIV-TRACE) (https://hivtrace.datamonkey.org/hivtrace, accessed on 16 August 2021) and MicrobeTrace (http://github.com/cdcgov/microbetrace, accessed on 16 August 2021) platforms were designed to rapidly predict and respond to large cluster outbreaks [19,20].

In Quebec, the provincial HIV drug resistance testing program, instituted in 2002, recommended universal baseline genotyping for all newly infected persons with primary (under 6 months) or chronic untreated infection. Our studies have applied population-level phylodynamics to characterize evolving trends in HIV-1 transmission over 18 years. Our findings have highlighted the contribution of primary/recent stage infections, I changed the frequently undiagnosed, in forward transmission of HIV spread among MSM [4,13,18,21,22,23]. This has contributed to the episodic development of large cluster networks (6+ members/cluster) that have fuelled the spread of HIV amongst MSM, rising from 13%, 25%, to 42% annual infections from 2004–2007, 2008–2011, and 2012–2015 periods, respectively.

In this study, we applied MEGA10, HIV-TRACE and Microbe TRACE molecular analyses to gain a more granular understanding of transmission clustering patterns amongst MSM and HET populations in Quebec over an 18-year period (2002–2020). Phylogenetic and available epidemiological data were combined to describe underlying factors sustaining onward spread of HIV-1 in the province and the impact of treatment-as-prevention initiatives.

## 2. Methods

### 2.1. Study Design and Populations

The HIV/AIDS epidemic in Quebec includes a concentrated domestic epidemic among MSM, established in the early 1980s, a largely historic subtype B epidemic among People Who Inject Drugs, and HET/MSM epidemics introduced through migration from countries where HIV is endemic. The provincial drug resistance testing program has accrued sequence datasets from 10,945 of the estimated 19,870 (15,940–23,800) persons living with HIV-1 in Quebec [24]. We applied molecular network analyses to elucidate evolving patterns of HIV transmission linkage in MSM (male only) and HET (female/mixed gender) subgroups harbouring B and non-B subtype infections. Test requisitions with non-nominative patient identifiers provided epidemiological information, including subject sampling date, date of birth, age, sex, viral load, stage of infection, treatment status and regional demographics.

Viral phylogenetic networks were built using Maximum Likelihood, Neighbour-joining (NJ) models within the Molecular Evolutionary Genetics Analysis software, version 7 and 10 (MEGA7/10) [18]. Viral sequences, spanning the protease and reverse transcriptase domains (HXB2 nucleotide positions 2262→3290 and 2253→3749), were aligned to consensus HXB2 sequences, removing gaps and cutting to identical sequence lengths (1497 base pairs) using BioEdit. Subtype B and non-B subtype trees were rooted against subtype K consensus sequence. Transmission clustering was assigned based on high bootstrap support (generally >95%) and short genetic distance (<1.5%).

Each newly genotyped person was assigned a non-nominative phylogenetic identifier based on HIV-1 subtype, cluster group membership, sex, and disease stage at first presentation. A viral lineage was defined as a group of viruses belonging to a shared phylogenetic cluster or a unique solitary transmission. Overall, there were 19,821 genotypes from 10.945 individuals, using birth dates to identify repeat subject sampling. Longitudinal sampling of individuals over the 18-year period revealed that nearly all persons were infected by a single founder viral variant that persisted over time. Several bad sequences (high sequence ambiguity) identified in phylogenetic trees were excluded from subsequent network analyses.

The phylogenetic-based identifier assigned to a given individual at first genotype remained invariant over time. As phylogenetic clustering was updated in real time (every year), patients retained their original cluster assignment and identifier, and first genotypes that fell into an existing cluster were considered as belonging to the same transmission cluster, infected with the same viral lineage, regardless of genetic distance. This phylodynamic tracking approach allowed for monitoring cluster evolution (cluster size, size distribution, and growth rate) in newly infected persons first genotyped over the 2002 and 2020 periods.

Individuals were segregated into three risk groups based on sex and putative cluster group: (i) the subtype B predominant MSM epidemic (66% of cases, *n* = 5024) was inferred, including male singleton transmissions (having no linked partnerships) and male-male clusters; (ii) the subtype B HET epidemic (*n* = 1345) included female singleton transmissions and mixed gender clusters; and (iii) the non-B subtype epidemics (*n* = 1848) included persons harbouring non-B subtype infections. Non-B subtype clusters having 6 or more male-only members inferred domestic crossover into the MSM population.

Evolving patterns of HIV transmission was ascertained in newly infected, treatment-naïve persons over the last 18 years. Individuals who were first genotyped following chronic-treatment failure (*n* = 2727) reflected infections arising prior to 2002 and were excluded from analyses of transmission dynamics. Treated persons within small and large MSM clusters (*n* = 339/3518) were, however, included in deducing epidemiological factors implicated in cluster growth and persistence.

#### HIV-TRACE and Microbe Trace Network Construction

Molecular transmission networks of cluster members established based on NJ-MEGA10 phylogenetic tree analyses were reconstructed and visualized using HIV-TRACE (https://hivtrace.datamonkey.org/hivtrace, accessed on 16 April 2021) and MicrobeTrace (http://github.com/cdcgov/microbetrace, accessed on 16 April 2021). Clustering patterns were assessed at genetic distance thresholds of 1.5% and 2.5%, using an ambiguity of 0.15 to resolve ambiguous nucleotides [19,20,25,26]. The genetic distance threshold of 1.5% is commonly used in research studies [25,27]. The longer 2.5% genetic distance threshold was included to describe the growth trajectories of several long-lived large cluster variants [25,26].

### 2.2. Epidemiological Correlates of Transmission Clustering

Individuals in predominant subtype B MSM, subtype B HET groups and non-B subtype groups were subdivided into three groups based on cluster size [14,15]. This included isolated singleton transmissions, small clusters (2–5 members/cluster) and large cluster networks (6–150 members/cluster). ANOVA (nonparametric) and contingency analysis ascertained epidemiological correlates of transmission clustering. Statistical analyses, frequency distributions, and heat maps were performed using GraphPad Prism software, version 9.00.

Sequence-based genetic diversity was used to assess the contribution of early-stage infection in the development and expansion of large cluster networks. Insofar as most HIV infections begin with a monophyletic founder variant, acquisition of mixed-based nucleotides in genotyped sequences served as a proxy marker of recency of infection [15]. A 0.44% mixed base cut-off identified early infection (under 6 months) based on longitudinal follow-up of PHI cohort participants with known estimated dates of infection. This cut-off threshold was consistent with other reported studies [28,29,30,31].

## 3. Results

### 3.1. Molecular Network Analysis of the HIV-1 Epidemics in Quebec

We applied Neighbour-joining phylogenetic methods within MEGA10 software to elucidate evolving trends in the spread of subtype B and non-B subtype epidemics among MSM and HET groups in Quebec over the last 18 years (2002–2020).

The *concentrated* subtype B epidemic among MSM (*n* = 5024, 79% of subtype B provincial cases, 2002–2020) showed high rates of transmission clustering (72%, male-only clusters) (Figure 1). Overall, half of the new infections were segregated as “dead-end” solitary transmissions (*n* = 1425) or short-lived small cluster networks (2–5 members/cluster, *n* = 891). The remaining half of the transmissions were associated with large cluster networks (*n* = 2708), with 6–157 members per clustered outbreak (mean cluster membership of 49) at the end of 2020.

The subtype B HET epidemic (21% of provincial subtype B infections) included 27% and 44% of transmissions within small (2–5 members) and large networks (6–49 members/cluster, averaging 23 members), respectively (Figure 1). Large cluster networks amongst HET groups reflected a historic epidemic spreading among PWID prior to 2010. Epidemic control among PWID in Quebec (*n* = 2 cases in 2017) was achieved through the early implementation of needle exchange and treatment programs. Notably, our findings differ from the rest of Canada, where PWID represented 17% of new cases in 2018 [12]. The subtype B HET epidemic, expanding over the last decade (2010–2020), was comprised of Quebecers originating from Haiti, the Caribbean, Mexico, Central and South America, and France [32].

The non-B subtype epidemic (*n* = 1947, 18% of the provincial epidemic) has arisen through migration from France and francophone countries in Africa and the Middle East. Overall, 2% and 11% of non-B subtypes are in large clusters reflecting domestic spread in HET and MSM groups, respectively (Figure 1).

### 3.2. Phylodynamics Reveal Shifting Trends in the Spread of the Subtype B Epidemic among MSM

Phylogenetics was used to infer longitudinal trends in onward spread of the subtype B infection in the MSM population over the 2002–2020 period, Phylodynamics revealed a 60% decline in singleton and small cluster subtype B transmissions over the 2008–2019 period (Figure 2A). This diminution reflects advances in anti-HIV treatment and prevention. Nevertheless, progress towards epidemic control amongst MSM has been offset by large cluster networks rising from 12% of new infections in 2002 to a steady 60% of new infections over the 2012–2020 period (Figure 2B).

Large cluster networks have played a pronounced role in the onward spread of HIV amongst MSM. Overall, 2.5% of large cluster lineages accounted for more than half of the epidemic in MSM. The median size of large cluster networks was 31 members (13–71, 25th–75th percentile range), averaging 49 members per cluster. The genesis, expansion, decay, and re-emergence of large cluster networks from 2002–2020 is shown in Figure 3. Over the 2015–2020 period, the growth of large cluster micro-epidemics have arisen through (i) expansion of pre-existent clusters, e.g., C163 (*n* = 151) and C027 (*n* = 77), adding 83 and 37 infections, respectively, over the five-year period; (ii) new clusters arising post-2014, e.g., C543R, C556, and C640 (*n* = 30, *n* = 18, and *n* = 12, respectively); (iii) demographic shifts in secondary clustered outbreaks, e.g., a C067 sub-epidemic ending in 2016 in Montreal (*n* = 45) followed by a Quebec City (*n* = 50) outbreak adding 24 infections since 2015; and (iv) secondary waves of drug-resistant sub-epidemics, the C543R and C159R epidemics, acquiring K103N, *n* = 20/30 and 14/46, respectively, since 2015.

The heatmap depiction of the growth trajectories of 44 large cluster micro-epidemics illustrates the complex interplay of virological, sociodemographic, and behavioural determinants in shaping HIV transmission dynamics (Figure 3). Although cluster sizes ranged from 20–157 members by the end of 2020, there was a median of seven new infections per year added (6–10 interquartile range) at peak periods of cluster growth, illustrating the pronounced role of early-stage infection in large cluster outbreaks. The persistence of large cluster micro-epidemics indicates a selective advantage of these super-spreader species. While most clusters were within the Greater Montreal Area, several clusters spread in Quebec City (labelled Q). Select clusters (labelled R) harboured K103N or G190A mutations, conferring resistance to first-generation non-nucleoside inhibitors.

### 3.3. Molecular Network Analyses Using HIV-TRACE and Microbe Trace

There have been debates as to which methodological approaches should be used to assign viral linkage, i.e., clustering, and reconstruct transmission networks [26,33,34,35,36]. We applied MEGA7/10 phylogenetic analyses across the viral reverse transcriptase (RT) and protease regions based on strict criteria of high bootstrap support and short genetic distance [34,35,36]. Our cluster group assignment was supported using multiple phylogenetic, genetic-distance, and Bayesian-based approaches [34,35,36]. Discordance among cluster group assignments frequently reflected the splitting of large clusters over time.

We applied HIV-TRACE and MicrobeTrace network analyses to gain a more granular understanding of the spread of the HIV epidemic in MSM. HIV-TRACE and MicrobeTrace network analyses were orders more rapid than NJ-MEGA10 phylogenetic analyses (days vs. weeks). Analyses could be performed at multiple genetic thresholds to identify clusters, monitor cluster growth, and forecast active transmission cluster networks.

As illustrated in Figure 4, HIV-TRACE topology of clustered male transmissions (*n* = 2+ male only clusters) at the commonly used genetic distance threshold of 1.5% depicts the disproportionate contribution of large cluster networks in sustaining the provincial epidemic amongst MSM (Figure 3). The Cluster 50 network (coloured in blue) includes 157 members by 2020, each harbouring the G190A resistance mutation. The Cluster 67 resolved into two clusters. The Montreal wave of the epidemic waned after 2016. The Quebec wave added nine new infections in 2017 with a cumulative acquisition of 20 infections from 2017–2020.

To assess the influence of genetic distance thresholds on cluster identification and outcomes, HIV-TRACE and MicrobeTrace clustering assignments were performed at 1.5% and 2.5% genetic thresholds using singleton, small cluster, and large cluster sequences pre-assigned based on MEGA10 phylogeny. As shown in Figure 5E, a genetic distance threshold of 2.5% was required to include all members of MEGA-10-defined large clusters. This relaxed threshold did not adversely lead to the coalescence of singleton transmissions or small cluster groups (Figure 5B,C).

The use of 1.5% genetic threshold led to many large cluster members being misclassified as singleton, dyad, or triad transmissions (Figure 4 and Figure 5D). The relaxed 2.5% criterion was required for large cluster super-spreader variants that persisted over extended time frames (Figure 3). This is underscored by the growth trajectory of cluster 67, assessed using Maximum-likelihood Neighbour-joining phylogeny (Figure 6A), MicrobeTrace (2.5% and 1.5%) (Figure 6B,C), and HIV-TRACE (1.5%) (Figure 4B) approaches. Demographic data showed initial outbreaks in Montreal followed by three secondary waves in Quebec City, Rimouski, and Chicoutimi (Figure 6D).

### 3.4. Epidemiological Features Implicated in the Growth Trajectories of Large Cluster Networks Fuelling MSM Epidemics

The frequency distribution of viral sequence diversity (% mixed base calls) substantiated the skewed contribution of early-stage infections in the episodic formation and expansion of large cluster networks (Figure 6A). Among MSM, 54.2%, 43.5%, and 27.6% of infections within large cluster, small cluster, and singleton groups were early-stage infections, respectively. Viral quasispecies diversity, indicative of early-stage infection, was significantly lower for large cluster micro-epidemics as compared to small clusters and singleton transmissions (Kruskal-Wallis statistic 523, *p* < 0.0001, post-hoc Dunn’s multiple comparisons tests) (Figure 6B).

Notably, the decline of singleton transmissions over the 2006–2017 period reflected a decline in the contribution of late-stage infections in transmission dynamics. Among singleton transmissions in MSM, early-stage infections (under 0.44% diversity) rose from 25%, 31%, and 41% infections in 2002–2007, 2008–2013, and 2014–2020, respectively. Median genetic diversity was significantly lower in 2014–2020 than the two earlier periods (Figure 7C, Kruskal-Wallis statistic 18, **** *p* < 0.001, post-hoc Dunn’s test ** <0.01 and **** *p* < 0.001, respectively).

Molecular network analysis revealed that large clusters were increasingly populated by younger men (Figure 8). The median ages (25th–75th percentile range) for individuals in singleton, small, and large cluster groups were 43 (35–50), 40 (32–48), and 36 (29–45) years, respectively (Kruskal-Wallis statistic 196, *p* < 0.001 post-hoc Dunn’s tests). Overall, persons under 30 years of age represented 14.8%, 20.5%, and 29.8% of infections in the singleton, small, and large cluster groups. Kernel density distribution showed that subjects within singleton transmissions and small clusters showed single peak curves with medians of 43 and 39 years of age, respectively. Conversely, the kernel density plots for large cluster networks were bimodal, with a median of 27 years of age and the other at 43 years of age (Figure 8).

Overall, regression analyses revealed that cluster size (1–150 members) was inversely correlated with viral genetic diversity (Pearson coefficient = −1894, *p* < 0.0001) and age of subjects (Pearson coefficient = −2037, *p* < 0.0001). Although large clusters showed significantly higher median viral loads than singletons (4.702 vs. 4.583 log copies/mL, *p* < 0.001), regression analysis did not show a correlation of cluster size with viral load.

### 3.5. Phylogenetic Inferences on the Spread of Non-Subtype B Epidemics

Recent migrants and asylum seekers represented 15–20% of newly declared HIV infections in Quebec from 2004 and 2015 and rose to 54% of new cases since 2017 [4,24,32]. The HET non-B subtype epidemic (16% of cases) included individuals arriving from francophone countries in Central Africa (subtypes C, A, D, and F), West & North Africa/France (subtypes CRF02_AG, CRF_06 and G), and Southeast Asia/East Africa (A/CRF01_AE) [4].

Phylogenetic network analysis revealed limited domestic dissemination of non-B subtype HET epidemics. The domestic spread of heterosexual clusters (3% of cases) included subtype D (*n* = 17), F1 (*n* = 12), F2 (*n* = 8), A1 (*n* = 9), and CRF01_AE (*n* = 6) large clusters (Figure 9).

Overall, 38% of HIV uninfected MSM screened at the community-based testing site in Montreal (2009–2013) included recent arrivals to Quebec (median 7 years in the province). The Montreal Primary HIV infection study cohort revealed the potential acquisition and spread of B and non-B subtypes among recent migrants, representing 26% of recruited MSM (*n* = 126/492). Overall, 44% acquiring subtype B infections belonged to large MSM networks (*n* = 56, 6–150 members, median cluster size 38). In contrast, phylogenetic analysis indicates that domestic crossover of non-B subtypes into MSM populations (*n* = 198) remains limited. As shown in Figure 8, several non-B subtype MSM outbreaks included two CRF01_AE clusters (*n* = 21 and 8), five CRF02_AG clusters (*n* = 6, 6, 10, 11, and 13) and a subtype A/B recombinant founder outbreak (*n* = 45).

### 3.6. Phylogenetic Inferences of Cluster Dynamics Using Integrase Sequences

Integrase strand transfer inhibitors-based regimens, introduced in 2009, are the recommended treatment of choice in HIV-1 management since 2013. The shift to integrase inhibitors allowed for comparative analyses of integrase sequence datasets. Cluster group membership for the integrase paralleled that which was observed for the RT and protease domains. Of note, clusters 45, 118 and 185 (*n* = 58, 64, and 54 members) shared a common integrase domain. These initial findings suggest the natural selection towards viral variants with RT, protease, and integrase enzymes render HIV-1 more transmissible or replicatively competent.

## 4. Discussion

In the mindset of HIV prevention, we used a large provincial repository of HIV-1 datasets accrued over 18 years to identify patterns of HIV spread in Quebec. We combined MEGA10 phylogenetic, HIV-TRACE, and MicrobeTrace network analyses to gain a more granular understanding of transmission evolving patterns of HIV transmission amongst MSM and HET populations. We have observed marked declines (~70%) in subtype B epidemics amongst MSM and HET groups from 2006–2020, concomitant to advances in treatment-as-prevention paradigms. Nevertheless, we have found that large cluster outbreaks and the cryptic spread of HIV related to recent migration remain hurdles to epidemic control.

The curbing of subtype B epidemics among MSM and PWID over the last 18 years reflect successes in treatment-as-prevention interventions in achieving population-level viral suppression, thereby reducing HIV transmissibility (i.e., undetectable = untransmissible) [37]. Several milestones include the introduction of potent simplified regimens (2007), first- and second-generation integrase inhibitors (2009 and 2015), pre-exposure prophylaxis (2013), and test-treat paradigms (post-2014). Of interest, epidemic control of epidemics among PWID and First Nation populations in Quebec contrasts to that observed in western Canada (2% vs. 47% of heterosexual cases, respectively) [11,32].

Overall, global control of epidemics among MSM has stalled [2]. Our studies have applied phylogenetic strategies to underscore the role of unchecked large cluster networks in the sustained growth of the epidemic among MSM in Quebec [15]. Our studies have found no evidence to support a correlation between large cluster size and individual-level sexual risk behaviour [15,38]. The size and duration of large cluster outbreaks are fuelled by transmission cascades of early-stage infections, spreading unevenly in younger populations. Poor testing habits and limited PreP uptake among MSM may contribute to the extended duration of large clusters [15,39]. Notably, infrequent testing was inversely associated with the number of reported sexual partnerships [15,39].

Cluster size is not a static measure but one that evolves with new transmissions, and the dynamics that drive large clusters are highly complex and may vary from cluster to cluster. Episodic risk dynamics can influence both the size and duration of acute infection outbreaks. As the numbers of HIV-infected persons taking life-long antiretroviral therapy expands, it remains imperative to achieve and sustain viral suppression to avert HIV transmission during chronic stage infection. The Swiss HIV study cohort group estimated that at least 14% of chronic stage transmissions occurred after treatment interruption [40].

Large cluster networks may also reflect the natural selection favouring more transmissible, infectious, or replicatively fit viral species [33,41,42,43]. Overall, 2.5% of viral variants have contributed to over half of infections. Our studies suggest that large cluster viruses may reflect a shift to evolutionary fit variants showing lower genetic diversity, higher viral loads, and extended high viremia [33,41,42,43]. Analysis of the distinct genotypic and phenotypic features of super-spreader viral lineages is warranted.

Classical epidemiological analysis distinguishes transmission events by risk groups, e.g., MSM vs. HET groups. Phylogenetics can improve cluster detection and cluster-based risk correlates. Transmission clusters have a mean transmission rate eight times the US national average [44]. In many publications, epidemiological comparisons are binary, i.e., clustered or non-clustered transmissions. Our findings show “short-lived” small cluster lineages epidemiologically resemble solitary transmissions while being distinct from “super-spreader” larger cluster variants. Stable large cluster lineages have been recently shown to fuel the Rhode Island, North Carolina, and New York statewide epidemics and may contribute to more transmissible drug-resistant epidemics [16,27,33,41].

The DHHS-USA recommended that health departments use a strict 0.5% genetic distance (0.005 nucleotide substitutions per site) metric to increase focus on more recent and rapid transmission clusters (5 new diagnoses/12 months) [44]. This shorter genetic distance is estimated to evolve over approximately 2–3 years [44]. Our findings found that this lower cut-off may not always assist in forecasting cluster growth [26,27,33]. Analyses at genetic distances of 1.5% and 2.5% thresholds assisted in distinguishing patterns of spread of large cluster networks implicated in the Quebec epidemic.

Recent migration has played a significant role in shifting patterns of viral spread in Quebec. The introduction of subtype B variants has arisen through migration from the Caribbean, Central and South America, France, and North Africa. There was a reported 2% seroprevalence of HIV among 156,000 Haitian Quebecers arriving from 2002–2010 [45]. Although onward transmission among HET migrant groups remains limited, there has been an increase in the acquisition and spread of HIV in recent MSM migrants associated with domestic subtype B and non-B subtype large cluster networks.

Molecular cluster detection is dependent on the timeliness and completeness of the sequence data included in the analysis. From the outset of the provincial genotyping program in 2002, universal genotyping was recommended for all newly infected persons. Although this allowed for in-depth population sampling (*n* = 10.945 persons), incomplete sampling could underestimate the number of active clusters. It remains important to genotype all individuals prior to treatment initiation (VL > 50 copies). Data from 2019 and 2020 may reflect incomplete sampling related to the COVID-19 epidemic.

There remains a clear need to understand how clusters translate to events that are actionable in public health. The cascade of undiagnosed early-stage infections yields viruses with low genetic diversity capable of overcoming severe transmission bottlenecks. Clearly, there remains a need to lower barriers to HIV testing. Only 8.3% of those eligible (*n* = 1,960,000) for PreP have accessed treatment in Canada [46]. In Quebec, new infections from migrant populations have risen to 50% of infections in 2017. Molecular surveillance can assist in the rapid identification of growing clusters to effectively direct and prioritize limited public health resources for younger persons and new migrant populations.

## 5. Conclusions

Determining the drivers of individual epidemics is critical in tailoring prevention measures to key vulnerable populations in a timely and effective manner. In this issue of Viruses, our study alongside that of Parks et al. shows how phylogenetics can assist in unravelling an in-depth sampled epidemic spreading among MSM, PWID, and HET groups in Quebec [32]. Nduva et al. and Bbosa et al. provide excellent studies describing the role of phylogenetics in describing viral spread in different regions of Africa [5,47]. For epidemic control in the next decade, there is a need to develop improved strategies to combine phylogenetic and epidemiological data to address and control for missed sampling [38]. Clustering and cluster size are not static measures, influenced by an array of virological, epidemiologic, and demographic factors.

## Figures and Tables

**Figure 1 viruses-13-01643-f001:**
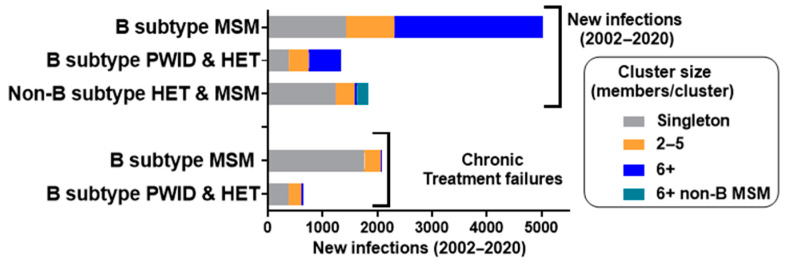
Phylogenetic resolution of the HIV epidemic in Quebec (2002–2020). Neighbour-joining MEGA10 phylogeny inferred singleton transmissions (grey), small-clustered networks with 2–5 members (orange), and large-clustered networks with 6+ members (blue). Newly genotyped, treatment-naïve persons were stratified into predominant MSM (male only) and subtype B PWID/HET (mixed gender) groups. Non-B subtype infections, introduced through migration, included HET and MSM, with 6+ male clusters depicted in green. The chronic treatment failure groups, often infected prior to 2002, did not co-cluster with newly diagnosed, drug-naïve persons (2002–2020).

**Figure 2 viruses-13-01643-f002:**
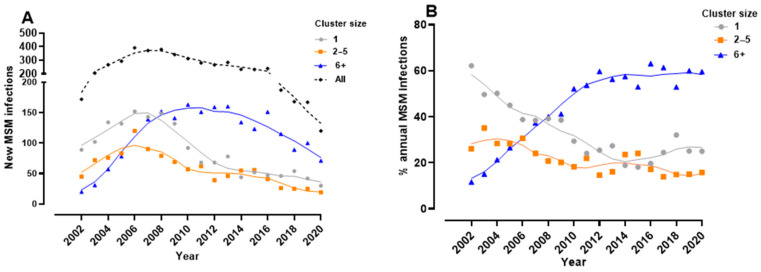
Phylogenetic resolution of patterns of spread of the subtype B epidemic amongst MSM. (**A**) The annual number of newly genotyped, treatment-naïve MSM segregating as singleton (grey), small-clustered networks (2-5, orange), or large-clustered cluster networks (6+ members, blue). Cluster group sizes of newly genotyped persons were assigned based on cumulative cases at end of each year; (**B**) The proportionate contribution of singleton transmissions (grey), small (orange), and large cluster (blue) networks in the annual growth of the subtype B HIV epidemic in the MSM population.

**Figure 3 viruses-13-01643-f003:**
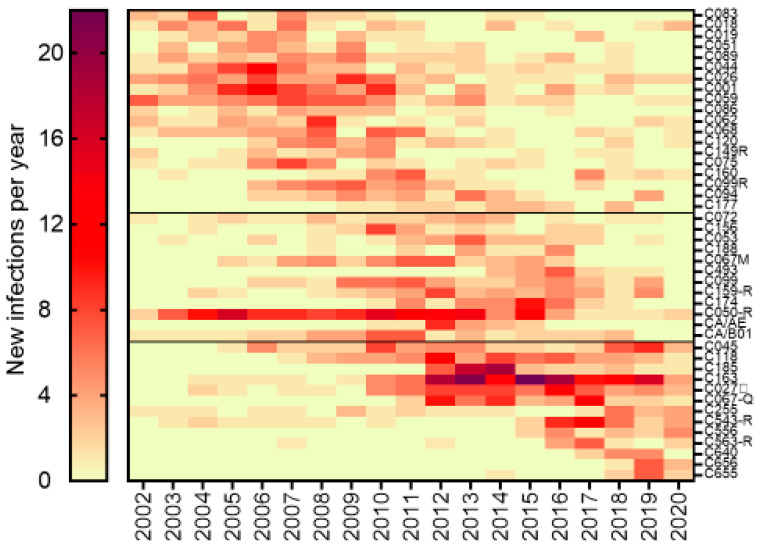
Growth trajectories of 44 large cluster MSM micro-epidemics having 20+ members. The heatmap colour gradient shows the number of infections added each year over the 2002–2020 period. Clusters were ordered chronologically to show waning, actively growing, and newly emergent clusters. The clusters harbouring drug resistance are indicated with an R.

**Figure 4 viruses-13-01643-f004:**
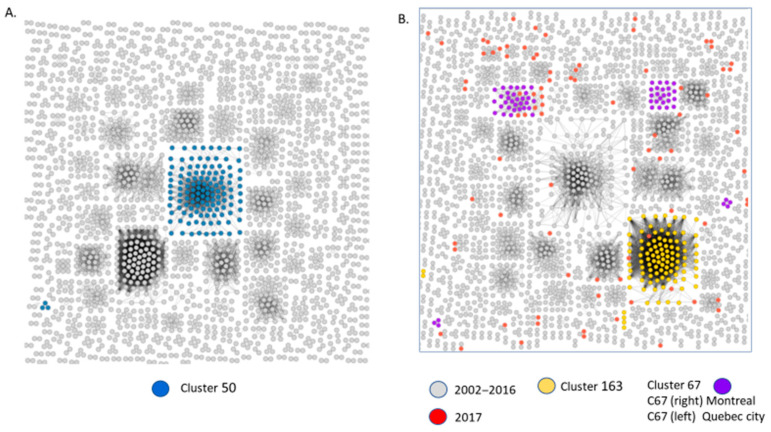
HIV-Trace resolution of clustered MSM networks (2002–2017). Network inferences depict clusters having 2 or more linked sequences at a 1.5% genetic distance. (**A**) Cluster 50 members (*n* = 142), coloured in blue, segregated as a large cluster (*n* = 125) and a triad (*n* = 3). (**B**) New infections added in 2017, coloured in red, were added to pre-existent clusters from 2002–2016, shown in grey; Cluster 163 (*n* = 122), depicted in yellow, increased by 5. Cluster 67 segregated into two clusters: a sub-epidemic circulating in Quebec City (left) and Montreal (right). Quebec City increased by 9 members in 2017.

**Figure 5 viruses-13-01643-f005:**
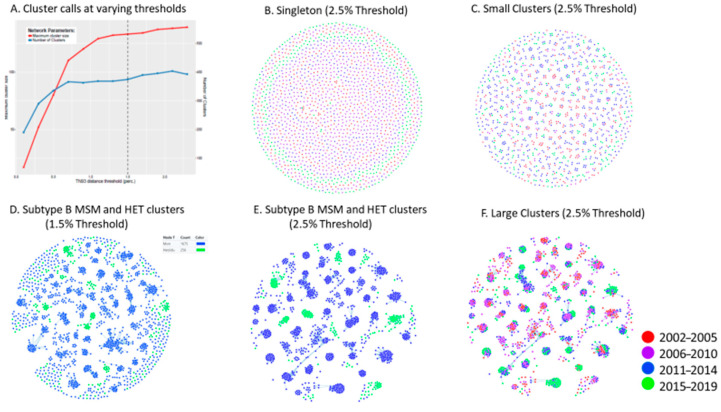
Phylogenetic resolution of patterns of HIV spread using HIV-TRACE and Microbe TRACE compared to MEGA10 phylogeny. (**A**) The number and size of individual clusters assigned by HIV-TRACE at different genetic distances (GDs). (**B**) MicrobeTrace topology (2.5% GD) of singleton transmissions assigned based on MEGA10 phylogeny. 5 (**C**) MicrobeTrace topology (2.5% GD) of small clusters assigned using MEGA10. (**D**,**E**) MicrobeTrace (1.5% and 2.5% GD) shows singleton/small cluster outliers at 1.5% failing to associate with MEGA10-defined large clusters. (**F**) MicrobeTrace (2.5% GD) depicts years of infection of large cluster members.

**Figure 6 viruses-13-01643-f006:**
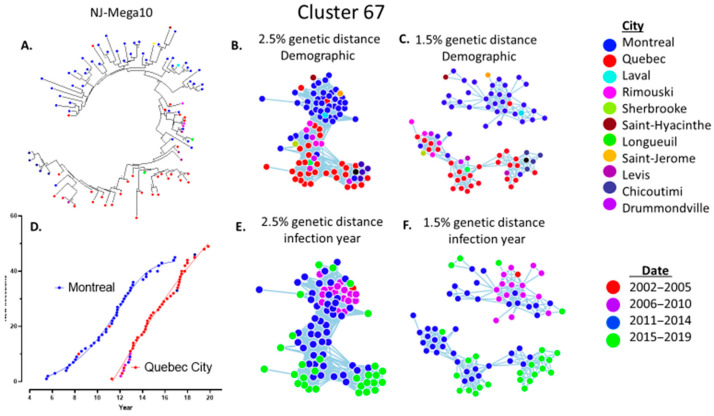
Phylogenetic topologies reveal recurrent waves implicated in the expansion and demographic spread of the cluster 67 sub-epidemic. (**A**) MEGA10 phylogeny of clade 67 reveals an initial wave in Montreal (blue nodes) followed by recurrent waves of spread in Quebec City (red nodes). (**B**,**C**) MicrobeTrace topology at 2.5% and 1.5% GD reveals demographic spread of clade 67 variants. (**D**) The alphanumerical year (2005–2020) of spread of clade 67 in Montreal and Quebec City outbreaks. (**E**,**F**) The quadrennial year of diagnosis of cluster 67 members.

**Figure 7 viruses-13-01643-f007:**
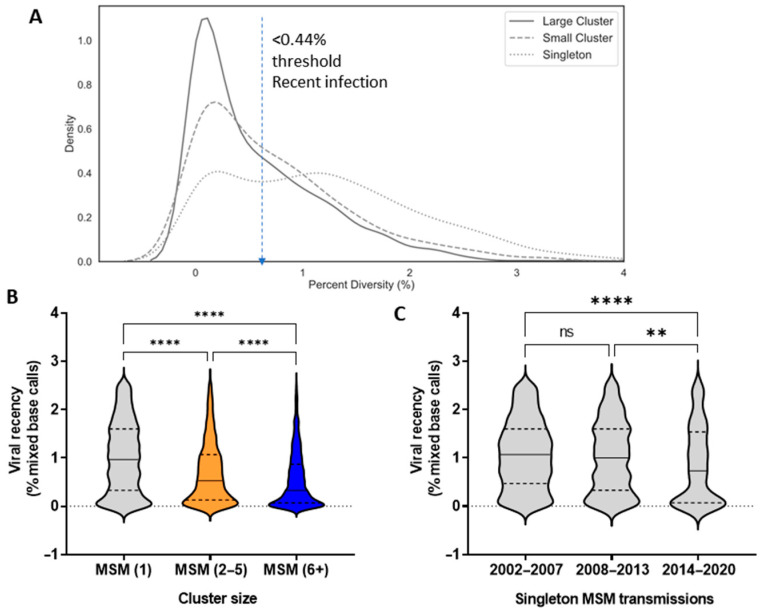
The pronounced role of early-stage infection in onward spread of HIV. (**A**) Frequency distribution of the baseline genetic diversity (% mixed base calls) of MSM infections segregating as singleton transmissions, small cluster (2–5 members), and large cluster (6+) networks. (**B**) Violin plots show the median viral diversity (25th–75th percentile range) of MSM infections within singleton, small cluster, and large cluster networks. (**C**) Changing trends in recency of infection for singleton transmissions arising over the 2002–2007, 2008–2013, and 2014–2019 periods.

**Figure 8 viruses-13-01643-f008:**
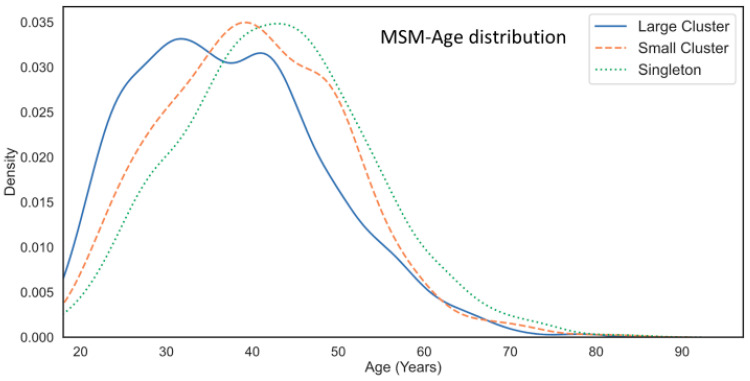
Age distribution of MSM participants in singleton, small, and large cluster networks. Kernel plots reveal the bimodal age distribution for large cluster networks.

**Figure 9 viruses-13-01643-f009:**
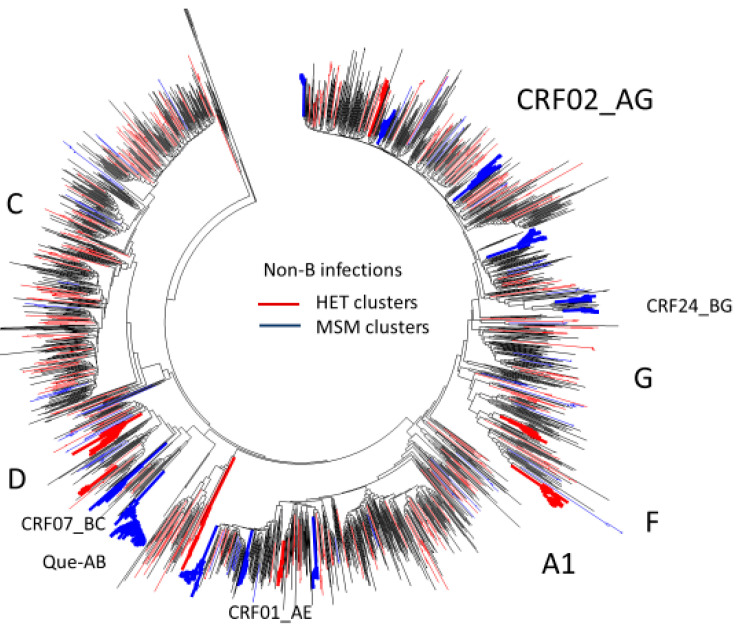
Phylogenetic resolution of the non-B subtype infections in Quebec (*n* = 1845). Neighbour-joining tree shows clustering in MSM (blue) and HET (red). Large cluster outbreaks reflect domestic crossover of non-B subtypes amongst MSM, including CRF02_AG, CRF024_BG, CRF01_AE, CRF07_BC, and a novel Quebec A/B founder variant.

## Data Availability

The Quebec HIV genotyping program sequences cannot be made publicly available for confidentiality reasons. An anonymized dataset of 233 sequences from PHI cohort participants belonging to the 30 largest transmission clusters is available on the Los Alamos HIV database website (GenBank accession numbers MK326905-MK327137). A representative portion of the tree has been previously published under GenBank accession numbers JF957375-JF957589.

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
