# Peer review of "The Role of Phylogenetics in Unravelling Patterns of HIV Transmission towards Epidemic Control: The Quebec Experience (2002–2020)"

_viruses, 2021, doi:10.3390/v13081643_

Round 1

Reviewer 1 Report

This paper used an extensive database of HIV-1 pole sequences to detect clusters of related HIV infections indicative of outbreaks or "superspreader" events in the Quebec region.  For most such studies journals should require that the data be made available to the public, but in this case it seems acceptable to withhold the data from the public because the study subjects were not counseled for human subjects data release. 

In figure 3, the legend and summary do not seem to match the colors in the cluster figure.  The summary mentions a "cluster 50" but the legend has a "cluster 163" as yellow and a "cluster 67" as blue, while the figure has some purple clusters but no blue cluster.  Figure 4 also mentions "cluster 67" so it would be nice to see it in figure 3.

Author Response

Comment 1 : In figure 3, the legend and summary do not seem to match the colors in the cluster figure.  The summary mentions a "cluster 50" but the legend has a "cluster 163" as yellow and a "cluster 67" as blue, while the figure has some purple clusters but no blue cluster.  Figure 4 also mentions "cluster 67" so it would be nice to see it in figure 3.

Reply: Figure 4 was revised. Figure 4A in shows cluster 50 in blue. Fig. 4B shows Cluster 67 in purple and cluster 163 in yellow.  

Editor Comment 1: Authors should provide reasoning for choosing the different genetic distance thresholds using the two clustering methods.

 Reply: Lines 291-313 text was added to provide reasoning for different genetic thresholds used with HIV TRACE and MicrobeTrace

 To assess the influence of genetic distance thresholds on cluster identification and outcomes, HIV-TRACE and MicrobeTRACE clustering assignments were performed at 1.5% and 2,5% genetic thresholds, using singleton, small cluster, and large cluster sequences pre-coded based on MEGA10 phylogeny. As shown in Figure 5E, a genetic distance threshold of 2.5% was required to include all members of MEGA-10 defined large clusters. This relaxed threshold did not adversely affect clusters within singleton and small cluster groups (Figure 5B and Figure 5C).      

 The use of 1.5% genetic threshold led to many large cluster members to misclassified as singleton, dyad, or triad transmissions (Figure 4 and 5D). The requirement of relaxed 2.5% criteria was required since large cluster super-spreader variants persist over an extended time frame (Figure 3). This is underscored by the growth trajectory of cluster 67 assessed using Maximum-likelihood Neighbour-joining phylogeny (Figure 6A), MicrobeTrace (2.5% and 1.5%) (Figure 6B and C), and HIV-TRACE (1.5%) (Figure 4B) approaches. Demographic data showed initial outbreaks in Montreal followed by three secondary waves in Quebec City, Rimouski, and Chicoutimi (Figure 6D). 

 Editor Comment 2: Authors indicate in text line 172 that Fig. 1 shows clusters identified for both HIVTRACE and MicrobeTrace but the legend only indicates the figures are for MicrobeTrace clusters.

Reply: Line 172 was deleted in revised text.

Editor comment 3:  Authors should compare and contrast the results of the three different clustering methods used.

 Reply: Lines 291-306 were added to compare and contrast methods.

Reviewer 2 Report

The manuscript by Brenner et al. presents a molecular analysis of the HIV epidemics of Quebec over the last 20 years. While the data are of interest for the reader of Viruses, there is place to improve the manuscript.

Major comments

  • Line 97-103. Rewrite the last § of the introduction by listing the objectives of the study rather than giving a summary of the conclusions. This § can be used as the first § of the discussion.
  • 10,945 sequences out of the almost 20,000 PLWH were analysed. Can this be a source of selection bias? It will be interesting to make it a “limitation paragraph” in the discussion.
  • Patient can be infected with two variants. Was it frequent in this series, if yes how was it handled?
  • Overall, the results section needs an extensive rewriting.
    • Line 169-174; Line 199-203; Line 208-236; Line 251-254; Line 298-300; Line 327-337. These § do not below to result section, they can be included in the discussion.
    • Figure 2; 4; 5; 6 Panel ABCD, … are missing in the figures.
    • The description of the result of the section 3.2 is rather short.
    • The text mentioned Fig 5C and Fig 5D (Line 278 and 289). There is only one panel in Figure 5.

Minor comments:

  • Line 66: For a complete picture of the situation, authors may add the new WHO’s target (95-95-95)
  • Line 153-159. Can the authors add a reference to support the use of two thresholds?
  • Can the authors use the same color code throughout the manuscript. Example Fig 2; 6 and 7.
  • Can the quality of the panel C of the figure 6 be improved?
  • Line 313. Clusters among the HET are limited to conjugal partners and family members. What are the hypothesis of the route of transmission for “family members”?
  • Line 315. Figure 8 instead of Figure 2
  • Line 349-350. How can TASP interventions reduce community viral load? This sounds to me as a shortcut.
  • Line 350. viRal

Round 2

Reviewer 2 Report

Minor comments:

  • Fig 1 caption: "Chronic treatment failure, virally suppressed individuals, frequently0 infected prior to 2002". please revised, the sentence has no sense.
  • Add title to figure 2, 5, 6.
  • The legends in the figure 1 & 2 are pixelized. please improve

Author Response

Minor comments:

  1. Fig 1 caption: "Chronic treatment failure, virally suppressed individuals, frequently0 infected prior to 2002". please revised, the sentence has no sense.

The sentence in Figure 1 was modified.

Lines 192-193 The chronic treatment failure group, often infected prior to 2002, did not co-cluster with newly diagnosed, drug-naive persons (2002-2020).

  1. Add title to figure 2, 5, 6.

The figure legends were modified to add titles.

Line 225: Figure 2. Phylogenetic resolution of patterns of spread of the subtype B epidemic amongst MSM.

Line 307-313: Figure 5. Phylogenetic resolution of patterns of HIV spread using HIV-TRACE and Microbe TRACE compared to MEGA10 phylogeny. 5A. The number and size of individual clusters assigned by HIV-TRACE at different genetic distances (GDs). 5B. MicrobeTrace topology (2.5% GD) of singleton transmissions assigned based on MEGA10 phylogeny. 5C. MicrobeTrace topology (2.5% GD) of small clusters assigned using MEGA10. 5D and 5E. MicrobeTrace (1.5% and 2.5% GD) shows singleton/small cluster outliers at 1.5% failing to associate with MEGA10-defined large clusters. 5F. MicrobeTrace (2.5% GD) depicts years of infection of large cluster members.

Lines 315-320. Figure 6. Phylogenetic topologies reveal recurrent waves implicated in the expansion and demographic spread of the cluster 67 sub-epidemic. 6A. MEGA10 phylogeny of clade 67 reveals an initial wave in Montreal (blue nodes) followed by recurrent waves of spread in Quebec City (red nodes). 6B and 6C. MicrobeTrace topology at 2.5% and 1.5% GD reveals demographic spread of clade 67 variants. 6D. The alphanumerical year (2005-2020) of spread of clade 67 in Montreal and Quebec City outbreaks. 6E and 6F. The quadrennial year of diagnosis of cluster 67 members.

  1. The legends in the figure 1 & 2 are pixelized. please improve

The legend to Figure 1 (lines 187-193) was modified. Figure 1. Phylogenetic resolution of the HIV epidemic in Quebec (2002-2020). Neighbour-joining MEGA10 phylogeny inferred singleton transmissions (grey), small-clustered networks with 2-5 members (orange) and large-clustered networks with 6+ members (blue). Newly genotyped, treatment-naïve persons were stratified into predominant MSM (male only) and subtype B PWID/HET (mixed gender) groups. Non-B subtype infections introduced through migration included HET and MSM, with 6+ male clusters depicted in green. The chronic treatment failure group, often infected prior to 2002, did not co-cluster with newly diagnosed, drug-naive persons (2002-2020).

The legend to Figure 2 (lines 225-230) was modified

Figure 2. Phylogenetic resolution of patterns of spread of the subtype B epidemic amongst MSM. 2A. The annual number of newly genotyped, treatment-naïve MSM segregating as singleton (grey), small-clustered networks (orange) or large-clustered cluster networks (6+ members, blue). Cluster group sizes of newly genotyped persons were assigned based as cumulative cases at end of each year; 2B. The proportionate contribution of singleton transmissions (grey), small (orange) and large cluster (blue) networks in the annual growth of the subtype B HIV epidemic in the MSM population.